# Evaluation of family planning service delivery in Gondar city public health facilities, Northwest Ethiopia: A cross-sectional study

Sefiw Abay[1], Tsega Hagos[2], Endalkachew Dellie[2], Lake Yazachew[2], Getachew Teshale[2]*, Ayal Debie[2]

1 Univesity of Gondar comprehensive specialized hospital, Gondar, Ethiopia, 2 Department of health systems and policy, Institute of public health, College of medicine and health sciences, University of Gondar, Gondar, Ethiopia

* getateshale1221@gmail.com

## Abstract

### Introduction

Family planning program is low-cost and an effective way to lower maternalmortality by reducing the number of high-risk births. Despite the effectiveness of the program, availability of materials, equipment and trained healthcare providers were some of the challenges in sub-Saharan African countries including Ethiopia. Determining the implementation status and identifying gaps is the aim of this evaluation.

### Method

A facility-based cross-sectional evaluation design with mixed method approach was employed. Quantitative data was collected through the exit interview of 477 clients from March 25-April 25, 2020. The evaluation focused on three dimensions: availability, compliance, and acceptability with multiple data sources. The quantitative data were entered in to Epi-data version 4.6 and exported to SPSS version 25 for analysis. Multivariable logistic regression was done to determine factors associated with client satisfaction. The qualitative data were transcribed, translated and analyzed by using thematic analysis. The evaluation finding was computed and compared with the preset criteria for the final judgment.

### Result

The majority of the health care providers (69.8%) got family planning training in the past two years. Three health facilities (37.5%) had 24hrs electricity with backup generators whereas only 25% of the health facilities had functional piped water inside the service room. Only two (25%) health facilities had a separate room for family planning service and 37.5% of health facilities had national FP guidelines. The overall availability of required resources for family planning service at Gondar city public facilities were 62.1%. Only twenty one (26.3%) of health providers dressed based on dressing code of ethics and none of them had ID during our observation. The overall compliance level of health care providers during providing family planning services were 75.5%. About 53.9% of the clients were satisfied with family

**Funding:** The author(s) received no funding for this work.

**Competing interests:** The authors have declared that no competing interest.

**Abbreviations:** CPR, Contraceptive Prevalence Rate; EA, Evaluability Assessment; FGAE, Family Guidance Association of Ethiopia; FP, Family Planning; HCW, Health Care Worker; IEC, Information Education and Communication; SOP, Standard Operative Procedure; TFR, Total Fertility Rate; PFSA, pharmaceutical fund and supply agency; UNFPA, United Nation Population Fund; USAID, United States Agency for International Development; WHO, World Health Organization.

planning service provided at Gondar city public health facilities.—and–were significantly associated variables with client satisfaction.

## Conclusion

The overall implementation of family planning service in Gondar city public health facilities with the three evaluation dimensions were judged as fair based on pre-setting judgment matrix. It is better to improve the service through training of healthcare providers, distributed family planning guidelines to health facilities and shortening of waiting time for service.

## Introduction

Family planning (FP) is defined as the capability of individuals and couples to anticipate and attain their desired number of children, spacing and timing of their births [1]. FP service saves the lives of women and children and improves the quality of life for all. It is one of the best investments that can help to ensure the health and well-being of women, children, and communities [2].

Globally, 12% of married women are estimated to have unmet need for family planning in 2017. The level was higher in Africa (22%) compared to other regions, where the unmet need for family planning is estimated to below 10% for married women [3]. The number of women who are married and have unmet need for family planning is declined in Asia and Europe [4].

In Ethiopia, family planning services was started in 1966 by the family guidance association of Ethiopia; a Non-governmental Organization (NGO). In 1975, the Ethiopian government started integrating family planning with maternal and child health services [5]. After the adoption of the population policy in 1993 several stakeholders have been involved in family planning promotion [6]. Although nearly two-thirds of married women had desire to space their pregnancies and one-third had desire to limit (cease childbearing), In Ethiopia 27% women in rural and 15% in urban areas had unmeet need of family planning [6].

Studies conducted in Tanzania identified that the key obstacles for FP services were lack of widespread trained providers, coupled with lack of stable provider competency and confidence, lack of consistent supply of family planning methods, equipment, materials, space, knowledge/interest on the part of potential users and possible provider bias that favor provision of short-acting methods [7]. The other study in Jimma Zone public health facilities in 2019 showed that 54% of the clients were dissatisfied with the FP service. There was a shortage of necessary equipment, supplies and Information Education and Communication (IEC) materials. To some extents the absence of standard FP guidelines affects providers' compliance [8]. A community based cross-sectional study conducted in northeast Ethiopia showed that 23.5% of pregnant mothers witnessed their current pregnancy was unintended [9]. Another survey conducted on FP program in Ethiopia showed that 77% and 79% public health facilities did not have weighting scale and BP measurement tools respectively. But, most public health facilities (68%) had equipped with trained healthcare providers in family planning service [10].

Despite the improvements in FP utilization in Ethiopia, there are still areas that need improvement. The annual discontinuation rate of family planning by method was IUD 13.3%, implant 10.9%, injectable 38.3% and pills 70.1% [11]. The reasons for such high discontinuation rates and unmeet need of family planning were not clearly understood. As a result, evaluating the program and identifying the gaps is very important to improve the program. Therefore, the findings of this evaluation may give significant value for all program

stakeholders, service users and policymakers to make an informed decision, to redesign the program and to revise guidelines, manuals, and training modules.

## Program stakeholders

Prior to the development of the proposal, an extensive discussion was held with Gondar city health department maternal and child health officers and "Ipas-Ethiopia" to identify other key stakeholders of the program. Service providers at each public health facility, family planning users (women aged 15 to 49), religious leaders, Gondar City Health Office (GCHO), Amhara Regional Health Bureau (ARHB), Ethiopian Pharmaceutical Supply Agency (EPSA) and Minister of Health (MoH) were considered as key stakeholders. During the Evaluability Assessment (EA), the stakeholders were involved in the development of evaluation questions, objectives, indicators, and judgment criteria.

## Program logic model

According to Ethiopian Demographic Health Survey (EDHS) 2016 report 22% married women had unmet need for family planning and overall, only 60% of currently married women aged 15–49 were satisfied on the demand of family planning service [6].

Number of providers who were trained on FP, number of people who have received information about FP, number of clients counseled on FP, number of clients received FP services, and numbers of complete reports sent on time are the expected output of the program activities. Improved knowledge & health seeking behavior, increase client satisfaction, increased utilization of the FP services, and improved decision making & service quality are the intermediate outcomes, while reduction of maternal and child morbidity and mortality and reduce poverty are impacts of the program (Fig 1).

## Methods

### Evaluation area and period

This evaluation was conducted in Gondar city public health facilities from March 25-April 25, 2020. Gondar city is located in the central Gondar zone which is 720 km away from Addis Ababa (the capital city of Ethiopia) and 180km from Bahir Dar (the capital of the regional state). Gondar is bordered with Lake Tana; the largest lake in Ethiopia and Simian Mountains;

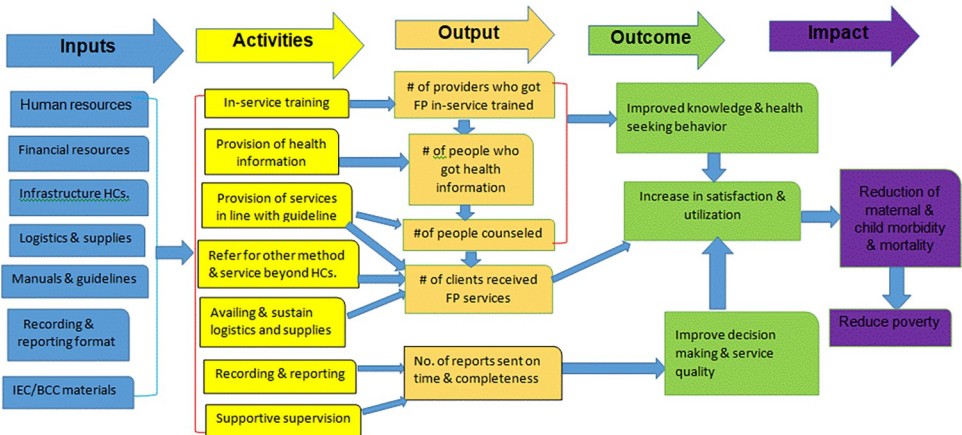

**Fig 1. Logic model of family planning service delivery in Gondar city administration public health facilities, Northwest Ethiopia, 2020.**

the highest mountain in the country. Based on the 2007/2015 national census conducted by the Central Statistical Agency (CSA), Gondar had a total population of 207,044; of this 108, 924 were women [12]. In Gondar city administrative there were eight public health centers and one referral hospital. All of the public and private health facilities offered family planning service.

## Evaluation approach and design

A facility based cross-sectional evaluation design with concurrent mixed method was employed. The qualitative data were used to support the quantitative findings. This evaluation mainly focused on the process theory of the program and includes availability, acceptability and compliance dimensions. A formative evaluation approach; an approach used to identify gaps and improve the program implementation was used in this evaluation. It is used to test the process theory of program or service.

## Sample size and sampling procedure

All health centers in Gondar city administration, all family planning service users (women aged 15–49), all family program managers, documents of family planning service and health care providers working in Gondar city health centers were the source population. Whereas, selected program managers, documents of family planning service, health care providers working in family planning, health centers and clients for family planning service during the study period were the study population. All health centers in Gondar city administration, clients who got family planning service at Gondar city health centers, service registers, program managers and health care providers working more than six months in the facility were the study units.

To measure the client's level of satisfaction the sample size was calculated by using single population formula, with 95% Confidence Interval (CI) and 5% margin of error and population proportion 66.1 (from a previous study conducted in Bahir Dar city) [13]. Thus, the sample size was calculated as n = $[Z^{a/2} P (1-P)]/E^2$ Where: n: Sample size, P: proportion of satisfied family planning clients, E: margin of error, $Z^{a/2}$: standard normal variable at 95% CI. By adding 10% non-respondents rate the final sample size of this evaluation was 380. To determine the sample size of the associated factors, double population proportion formula was used for variables such as education, age of clients, and privacy [14]. For all variables, the sample size was calculated considering 80% power, 1 to 1 ratio, and 10% non-response rate which yield 477 samples. Since, the sample size calculated for the double population was greater than that of the single population proportion 477 was the final sample size used in this evaluation. Finally this sample size were distributed proportionally to all public health centers at Gondar city based on the number of clients served in the last six months in each health facilities. Each study participants (family planning service users) were selected with simple random sampling technique.

The availability of required resources was observed in the eight public health facilities by using standardized resource inventory checklist. The sample sizes for qualitative data (key informant interview) were determined by information saturation and the samples were selected purposively (Maternal and Child health (MCH) coordinators, program managers and senior health care providers (midwives).

## Variables and measurements

In this evaluation client satisfaction was the dependent variable whereas, distance from health facilities, waiting time, waiting area, cleanness of the service room, the tendency to have a

child, plan when to have a child, accessibility & availability of logistics, equipment and supplies, client-provider interaction, socio-demographic variables of clients (age, education status, marital status, occupation, religion, income, family size and place of residence), availability of trained and technical competent FP provider [7] and availability of IEC materials [8] and guidelines were independent variables. All the independent variables were identified from literature review and introduction section of this evaluation work and the data related with these variables were collected through exit interview and observation.

### Availability of resources

Physical existence and functionality of resources needed for family planning services. It was measured through observation by using inventory checklists [15,16] with nine indicators and weighted value of 30%.

### Compliance

It is the adherence of health care providers to FP service guidelines [17] and measured through client-provider interaction and document review with 11 indicators. It was measured through observation by using observation checklist [16] and given weighted value of 40%.

### Acceptability

It was measured as client satisfaction and measured by five point likert scale of 13 items interviewer administered questionnaires [3]. The final satisfaction interviewed answers were dichotomized as satisfied and dissatisfied by using demarcation formula: ((Max-Min)/2) +Min. Based on this those clients who scored above 37 were considered as satisfied and below were dissatisfied [3,18]. It had 13 indicators with weighted value of 30%.

### Overall FP service implementation

It was measured by using 33 indicators for the three dimensions: availability, compliance and acceptability. The implementation status of the program was judged as poor (<59%), fair (60–74%), good (75–84) and very good (>85%) based on pre-setting judgment cut-off point. The indicators were adapted from the objective and strategies of national family planning guidelines and other relevant readings [10,19,20].

### Data collection procedures and quality control

Quantitative and qualitative data were collected by using validated tools adapted from different literatures [21–24]. Accordingly, structured questioners for exit interview and resources inventory checklists for availability observation were used. Exit interview questionnaires were transcribed, translated into Amharic and re-translated back to English version to ensure its consistency.

To minimize bias during observation, the observer discarded the first and the last three observations [20,25]. Two days training were given for the data collectors and supervisors. Pre-test was done among 24 participants (5% of the total sample size) in the health facilities outside the evaluation area. The principal investigator together with supervisors checked the completeness and consistency of the collected data every day.

### Data management and analysis

The quantitative data were coded, entered to Epi data version 4.6 and exported to SPSS version 25.0 for analysis. Data collected from resource inventory were entered in to Microsoft Excel 2013 for further analysis. To identify variables affecting the satisfaction of clients, binary and

multivariable logistic regression was done. In the final model variables having p- value of ≤0.05 with 95% CI were considered as predictors. Descriptive findings were presented using tables, graphs and charts.

Qualitative data were recorded, noted, transcribed in Amharic, translated to English and coded manually. Finally themes were created according to the evaluation dimensions: availability related, compliance related and satisfaction related with the help of open code software. The qualitative findings were presented in each dimension to explain and support the quantitative findings.

## Results

A total of 477 study participants for exit interview with a response rate of 100%, 16 key informant interviews (2 in each health facilities), 80 observations (10 in each health facilities) and three months retrospective document review were done for this evaluation.

### Availability of resources

In Gondar city public health facilities (health centers) there were a total of 133 health care workers (84 nurses, 28 midwives and 21 public health professionals). All public health facilities included in this evaluation had two or more health care providers assigned to provide family planning service. Only 42 (31.2%) of healthcare providers working in the facilities had training on family planning. Half of the health facilities (50%) had functional blood pressure apparatus and lamp light, 37.5% had functional weight scale, and 28.6% had pregnancy test kits inside the service room. Majority of the health facilities (87.5%) had functional autoclave and functional minor surgery equipment (87.5%) but only one health facility had sterile surgical drapes (Table 1).

Key informant interview results also revealed that turnover of trained health care providers and shortage of materials challenged them to give quality family planning service.

"Trained health care workers are frequently left our health facility and transferred to other facilities. Due to that, the service is given by fresh graduate midwives and nurses. In addition all the necessary equipments are not available all the time. All these issues may compromise the quality of the service."

[36 years old female health center director]

**Table 1. Equipment and supplies available for family planning service delivery at Gondar city public health facilities, Northwest Ethiopia, 2020.**

| | Available in the HFs at the time of data Collect. | | |
|---|---|---|---|
| Equipment (Reusable) | functional during observation | (n = 8) | Availability in % |
| Functional Sterilizer (Autoclave) in the HC | ✓ | 7 | 87.5% |
| Functional Blood pressure apparatus | ✓ | 4 | 50% |
| Stethoscope | ✓ | 3 | 37.5% |
| Functional Thermometer | ✓ | 2 | 25% |
| Functional Weight Scale | ✓ | 3 | 37.5% |
| Clean Instrument trays | ✓ | 5 | 62.5% |
| Examination couch or table? | ✓ | 7 | 87.5% |
| Functional equipment lamp light | ✓ | 4 | 50% |
| Functional uterine sound? | ✓ | 6 | 75% |
| Functional Speculum | ✓ | 7 | 87.5% |
| Functional Scissors | ✓ | 7 | 87.5% |
| Functional Temecula | ✓ | 6 | 75% |
| Minor Surgery equipment's | ✓ | 7 | 87.5% |
| **Average Index** | | | **65.4%** |

Only three HFs (37.5%) had standard FP guidelines revised in 2015 and seven of HFs (87.5%) had standard referral form and recording logbook. Although most resource required for FP service are available in the health facilities, five (87.5%) of them had no pregnancy test kits and sterile surgical drapes. Most family planning methods; Implanon, Jadelle, combined oral contraceptives, combined injectable contraceptives (depo) and male condoms were available in all public health facilities during data collection. However, some family planning methods such as female condoms, progestin-only injectable contraceptives, spermicidal, cycle beads for standardized method, male and female sterilization were not given in all public health facilities. Six HFs (75%) had no separate physical examination room. Only some HFs had piped water inside the service room with soap and hand washing area (25%) and waiting area (37.5%). But none of them had a single-use towel, chair, entertainment media and telephone. All HFs had tables, toilet and 24 hour electricity power. Half of HFs (50%) had anatomical models and flip charts inside the FP service room. Only some HFs had information sheet (62.5%), brochure (37.5%), job aids (WHO medical eligible criteria wheel) (75%) and direction indicator of service area (87.5%) (Table 2). The key informant interviews also revealed that there is insufficiency of some resources for family planning service delivery.

"Although short term family planning (injectable and Depo) and progesterone only oral contraceptive were stocked out in the recent one to two weeks, there are sufficient amount of supplies, contraceptives, and equipment for family planning services. But family planning trainings is not adequately provided for all health professionals. There is also an imbalance between the number of users and the number of health professionals who are working in the family planning room."

[29 years old female BSc midwifery and MCH head]

"There is high turnover of trained health professionals, lack of room to put our functional and non-functional materials as well as difficulty to keep patient's privacy during consultation and examination are the challenges to provide family planning service."

[28 year old male BSc nurse and HC director]

**Table 2. Judgment matrix of availability dimension to provide family planning service in Gondar city administrative public health facilities, Northwest Ethiopia, 2020.**

| Dimension with indicators | Weight | Expected | Observed in no. | Achievement in % | Judgment | Judgment Criteria |
|---|---|---|---|---|---|---|
| **Availability (30%)** | | | | | | |
| Number of service providers trained on FP methods in the past 2 years. | 14 | 14 | 4 | 28.6% | Poor | [≥85] V. Good [75–84] Good [60–74] Fair [≤59] Poor |
| The proportion of health facilities having functional basic instruments/equipment to provide FP. | 15 | 15 | 9.8 | 65.4% | Fair | |
| The proportion of HFs having all FP contraceptives | 12 | 12 | 12.7 | 67.8% | Fair | |
| The proportion of HFs having supplies to provide FP. | 15 | 15 | 12.6 | 83% | Good | |
| The proportion of HFs having at least one updated FP guideline. | 13 | 13 | 6.5 | 50% | Poor | |
| The proportion of HFs having separate FP examination room | 10 | 10 | 2.5 | 25% | Poor | |
| The proportion of HFs having IEC material for FP. | 9 | 9 | 5 | 55.4% | Poor | |
| The proportion of HFs experience stock out of contraceptive for the last consecutive 3 months. | 6 | 6 | 4.1 | 67.8% | Fair | |
| Proportion of HFs having all recording & reporting formats. | 6 | 6 | 4.9 | 81.3% | Good | |
| Overall average score of availability dimension | 100 | 100 | 62.1% | | Fair | |

## Compliance of healthcare providers with the national guideline

Provision of counseling to new and repeat family planning users should be in line with the national guideline. Only twenty one (26.3%) of health providers dressed based on dressing code of ethics and had ID while during our observation; sixty four (80%) clients discussed with the health care providers about preferred FP method and only thirty seven (46.3%) clients got important basic information about each FP method. only twenty three (28.7%) healthcare providers had used guidelines consistently while giving the service. The overall compliance level of health care providers during providing family planning services were 75.5% (Table 3).

Participants of key informant interview also revealed that family planning service delivery are not usually based on the national guideline.

> "The health professionals didn't get adequate refreshment training. Some healthcare providers also confine the family planning to their religion and they didn't frequently use the guidelines. The guideline itself is not available in working room. Due to these and other factors the family planning service may not in line with the standards"

[24 years old female midwifery]

## Client satisfaction

**Socio-demographic characteristics of respondents.** Among all respondents 206 (43.2%) were mothers with breast-feeding at the time of data collection and 221(46.3%) were house wives. One fourth of the respondents 123 (25.8%) were illiterates and most of them were Orthodox Christian followers 415(87%). From the total respondents three hundred ninety-

**Table 3. Judgment Matrix of Compliance dimension of family planning service in Gondar city administrative public health facilities, Northwest Ethiopia, 2020.**

| Dimension with indicators<br>2. Compliance (40%) | Weight given | Expected | Observed | Scored | achievement in % | Judgment parameter | Judgment Criteria |
|---|---|---|---|---|---|---|---|
| Proportion of healthcare providers follow infection control procedures during service delivery | 10 | 10 | 5 | 5 | 51.2% | Poor | [≥85]<br>V. Good<br>[75–84]<br>Good<br>[60–74]<br>Fair<br>[≤59]<br>Poor |
| Proportion of Healthcare providers treat clients with respect | 9 | 9 | 8.4 | 8.4 | 93.8% | V. Good | |
| Proportion of HFs stored contraceptives properly (No exposure to rain and sun, protected from rats and pests). | 8 | 8 | 7.2 | 7.2 | 90% | V. Good | |
| Proportion of healthcare providers offered at least two modern methods of FP. | 7 | 7 | 7 | 7 | 100% | V. Good | |
| Proportion of HFs that follow the consistency of recording and reporting the last 3 months Performance. | 9 | 9 | 8.2 | 82 | 91.3% | V. Good | |
| Proportion of service providers asks the client about reproductive intentions. | 11 | 11 | 7.6 | 7.6 | 68.8% | Fair | |
| Proportion of service providers teach about HIV/AIDS during service provision. | 11 | 11 | 4.8 | 4.8 | 43.8% | Poor | |
| Proportion of clients participating actively in discussion and selection of methods | 9 | 9 | 7.9 | 7.9 | 87.5% | V. Good | |
| Proportion of healthcare providers demonstrated good counseling skills. | 8 | 8 | 5.8 | 5.8 | 72.5% | Fair | |
| Proportion of client got basic information on the selected method (Complications, side effects) | 10 | 10 | 6 | 6 | 61.3% | Fair | |
| Proportion of Service provider Gives instructions on when to return. | 8 | 8 | 7.6 | 7.6 | 95% | V. Good | |
| Overall average score of Compliance dimension | 100 | 100 | 75.5 | 75.5 | 75.5% | Good | |

NB: Scored = (Observed X Weight)/Expected, Achievement in percentage = (Scored/Weighted) X 100%.

eight (83.44%) were urban dwellers, 383 (80.3%) were married and the mean age of the respondents was 26. From the total respondents 396 (83.0%) were repeated family planning users and 298 (62.5%) had discussed with their husband about family planning they utilized (Table 4).

## Client satisfaction with the family planning service

The finding showed that the overall client satisfaction by family planning service was 53.9% with 95% CI. The level of satisfaction with the cleanness of family planning room was 212 (44.4%) and 235 (49.3%) clients were satisfied with the privacy during examination and consultation in the health facilities. Less than half of the clients 232 (48.6%) were satisfied with the overall service they had received (Table 5).

## Factors affecting client satisfaction on family planning service

Frequency of visit, waiting time to get the provider, discussion with husband about family planning, age of clients, marital status, main occupation, educational status, and household monthly income were significantly in binary logistic analysis with 95% CI.

In the multivariable logistic regression analysis, waiting time to get the provider, and household monthly income were predictor variables with 95%CI. Clients who waits greater than 30 minutes to get the service provider were 46% less likely satisfied than those who waited less than 30 minutes (AOR:0.54, 95% CI:0.34,0.79). Accordingly, clients whose household income is between 1001–2000 were 58% less likely satisfied than those whose household income is ≤500 (AOR: 0.42, 95% CI: 0.18, 1.54). and those clients with household income of ≥ 2001 were 55% less likely satisfied as compared with having household income of ≤500 (AOR: 0.45, 95% CI: 0.26,0.78) (Table 6).

## Judgment matrix

The overall implementation status of family planning program was 62.9% and judged as fair. Availability, compliance and acceptability of family planning program services were 62.1, 75.5 and 46.9% and judged as fair, good and poor as per the preset judgment parameter, respectively (Table 7).

## Discussion

The finding of this evaluation showed that most FP methods were available in the evaluation area during data collection and in the last three months. However, in some HFs, progestin-only oral contraceptives, female condoms, cycle beads for standard day's method and emergency oral contraceptive were stocked out during the past three months. This evaluation result is in line with country wide study in Ethiopian which was availability of FP is high. In the HFs, short acting FP methods were available in 100%. But in some facilities occasional stock out were experienced in the past three months [26]. In the current evaluation, only 31.2% (42) of the health care providers got in-service training in the past two years. In this evaluation, only 28.7% FP service providers use the guide line and protocols consistently. This finding was inconsistent with the WHO guideline [27]. This may be due to unavailability of guide lines, weak supportive supervision and unavailability of continues monitoring and evaluation system.

By this evaluation only three (37.5%) of the HFs had a copy of FMoH FP guideline. This finding was incongruent with study done in Jimma zone, only 50% had a copy of FMoH FP guideline [28]. According to FMoH the guideline has an objective to guide all health care providers directly or indirectly involved in the provision of FP services [29]. In case of availability

**Table 4. Socio-demographic characteristics of family planning service clients in Gondar city public health facilities, Northwest of Ethiopia, 2020.**

| Variables | | frequency (n = 477) | Percent (%) |
|---|---|---|---|
| Residency | Urban | 398 | 83.4 |
| | Rural | 79 | 16.6 |
| Age in year | 15–19 | 37 | 7.6 |
| | 20–24 | 175 | 36.7 |
| | 25–29 | 140 | 29.4 |
| | 30–34 | 71 | 14.9 |
| | ≥35 | 54 | 11.3 |
| Marital status | Single | 82 | 17.2 |
| | Married | 383 | 80.3 |
| | Divorced | 3 | 0.6 |
| | Widowed | 9 | 1.9 |
| Religion | Orthodox | 415 | 87.0 |
| | Muslim | 50 | 10.5 |
| | Protestant | 2 | 0.4 |
| | Catholic | 3 | 0.63 |
| | Jewish | 7 | 1.5 |
| Educational status | Cannot read & write | 123 | 25.8 |
| | Read & write | 37 | 7.8 |
| | Primary (1–8) | 118 | 24.7 |
| | Secondary (9–12) | 116 | 24.3 |
| | College and above | 83 | 17.4 |
| Main occupation | Not employ | 33 | 6.9 |
| | Government employ | 36 | 7.6 |
| | Merchant | 59 | 12.4 |
| | House wife | 221 | 46.3 |
| | Daily labor | 91 | 19.1 |
| | Student | 37 | 7.8 |
| No of children <15 years | Have no children | 118 | 24.7 |
| | 1–4 children | 348 | 73.0 |
| | >4 children | 11 | 2.3 |
| House hold income | ≤500 | 23 | 4.8 |
| | 501–1000 | 53 | 11.2 |
| | 1001–2000 | 137 | 28.7 |
| | 2001–3000 | 110 | 23.1 |
| | ≥3001 | 154 | 32.3 |
| Do you came to this HC before | No, I am new visiter | 396 | 83.0 |
| | Yes | 81 | 17 |
| Time home to HC in minutes | ≤30 | 298 | 62.5 |
| | 31–60 | 139 | 29.1 |
| | ≥61 | | 8.4 |
| Usually discussing with husband about FP | Yes | 150 | 62.9 |
| | No | 168 | 35.2 |
| | Not remember | 9 | 1.9 |
| Currently breast feeding | Yes | 206 | 43.2 |
| | No | 271 | 56.8 |

*(Continued)*

**Table 4.** (Continued)

| Variables | | frequency (n = 477) | Percent (%) |
|---|---|---|---|
| Plan for next children (among who need's child) | Immediately | 40 | 11.2 |
| | One-two year | 67 | 18.8 |
| | After two year | 219 | 58.5 |
| | I do not know | 31 | 8.7 |

of electricity and water supplies three (37.5%) of the HFs had functional light source with backup generator, whereas only two (25%) of the HFs had water in the room. Generally a situational analysis of FP conducted in Ethiopia reported that the service environment and infrastructure were good; but some was limited in health facilities including water supply [30].

In this evaluation only two (25%) HFs had separate examination room for provision of FP service. This finding seems similar to report of FMoH of Ethiopia which results in most HFs the space or room for the provision of family planning is integrated with other reproductive health programs [31].

During the counseling session, the use of IEC materials especially for illiterate clients helps to understand key information and helps the provider to remember important points [32]. During observation of this evaluation 72.5% of the family planning service providers utilize at least one or more IEC materials; predominately sample contraceptives were utilized to

**Table 5. Judgment matrix of acceptability (satisfaction) dimension of family planning service in Gondar city administrative public health facilities, Northwest Ethiopia, 2020.**

| Dimension with indicators | Expected | Weight given | Observed | Scored | Achievement in % | Judgment | Judgment criteria |
|---|---|---|---|---|---|---|---|
| **Satisfaction (30%)** | | | | | | | |
| Percentage of clients perceived the distance between from service area was convenient. | 7 | 7 | 3.4 | 3.4 | 48.4% | Poor | [≥85] V. Goo [75–84] Good [60–74] Fair [≤59] Poor |
| Proportion of clients satisfied with the cleanness of the family planning room. | 7 | 7 | 3.1 | 3.1 | 44.4% | Poor | |
| Proportion of clients perceived schedule or working hours of family planning service is appropriate. | 9 | 9 | 3.9 | 3.9 | 42.8% | Poor | |
| Proportion of clients satisfied with the convenience of the waiting room. | 10 | 10 | 3.3 | 3.3 | 33.1% | Poor | |
| Proportion of clients satisfied with the convenience of the counseling room. | 11 | 11 | 5.3 | 5.3 | 48.2% | Poor | |
| Proportion of clients who perceive examination and consultation they had today was satisfactory. | 8 | 8 | 3.9 | 3.9 | 49.3% | Poor | |
| Proportion of clients perceived their Privacy was maintained in the service room. | 7 | 7 | 3.5 | 3.5 | 49.9% | Poor | |
| Proportion of clients satisfied with the Provider's explanation about the contraceptive methods. | 9 | 9 | 4.3 | 4.3 | 47.6% | Poor | |
| Proportion of clients satisfied with the provider used teaching aids. | 7 | 7 | 2.5 | 2.5 | 35.2% | Poor | |
| Percentage of client satisfied with HFs having functional latrine & piped water. | 8 | 8 | 3.4 | 3.4 | 42.6% | Poor | |
| Proportion of clients satisfied with the overall service that they get. | 7 | 7 | 3.4 | 3.4 | 48.6% | Poor | |
| Proportion of clients perceived the provider's skills who gave the service for them was good. | 5 | 5 | 2.9 | 2.9 | 58.5% | Poor | |
| Proportion of clients satisfied with the appointment time. | 5 | 5 | 4 | 4 | 80% | Good | |
| **Overall average score of acceptability dimension** | **100** | **100** | **46.9** | **46.9** | | **Poor** | |

NB: Scored = (Observed X Weight)/Expected, Achievement in percentage = (Scored/Weighted) X 100%.

**Table 6. Bi-variable and multi-variable logistic regression analysis result of clients' satisfaction on service acceptability of family planning service in Gondar city public health facilities, Northwest Ethiopia, 2020.**

| Variables | Satisfaction | | | |
|---|---|---|---|---|
| | Yes Frequency (%) | No Frequency (%) | COR (95%CI) | AOR (95%CI) |
| House hold income | | | | |
| ≤500 | 15 (62.20) | 8 (34.80) | 1 | 1 |
| 501–1000 | 26 (62.20) | 27 (50.90) | 0.51(1.86,1.41) | 0.76(0.27,2.164) |
| 1001–2000 | 64 (46.70) | 73 (53.30) | 0.47(0.86,1.17) | 0.42(0.21,0.85)* |
| ≥2001 | 152 (57.6) | 11 2(42.4) | 0.96(0.38,2.41) | 0.45(0.26,0.78)* |
| Waiting time | | | | |
| < = 30 | 205 (57.90) | 149 (42.10) | 1 | 1 |
| > = 31 | 52 (42.30) | 71 (57.70) | 0.53(0.35,0.81) | 0.54(0.35,0.84) * |

* Variables associated with the client satisfaction in multi-variable logistic regression analysis.

demonstrate good counseling skills. Certain information on family planning methods was considered as essential to aid the decision making process [29]. In this evaluation, 61.3% clients got basic information (side effects and complication) of the accepted method. This is almost similar with study conducted in Ethiopia which results three quarters (73%) were informed about how to use the contraceptive method [33].

According to WHO, for those methods that require surgical approaches, insertion, fitting and/or removal by a trained healthcare providers, appropriate infection prevention procedures must be followed [34]. Including hand washing practices before and after performing procedures was only 48.8% and only 43.8% service providers were discussed about HIV/AIDS during FP service delivery.

One principal determinant of continued utilization of family planning services is client satisfaction with the services [35]. The long waiting time for clients to get a service is one of the factors affecting implementation of FP service [35]. In this evaluation, only 3.3% of clients were waited for more than one hour to get the service provider which is higher than study report at Jimma zone 1.7% [36]. Likewise, about 74.2% of the client got service within the acceptable waiting time (30minutes). This result is higher than a comparative study findings on quality of family planning done in Ghana, Kenya, and Tanzania which was 42.1% [35] as well as a study conducted in Jigjiga town was 34.9% [33], But lower than a study result conducted at Jimma zone health centers (92.4%).The variation can be due to difference in facilities working culture, the number of health professionals assigning in the room, client flow and the recent reform implementation.

Among all evaluation participants who received family planning service, 134 (28.1%) had monthly household income of ETB of 1001–2000 and half of them (53.3%) were not satisfied with the family planning service. But clients whose monthly household income greater than

**Table 7. Overall Judgment matrix and analysis for family planning program evaluation at Gondar city administrative public health facilities, Northwest Ethiopia, 2020.**

| Dimension with indicator | Weight given | Expected | Observed score | Achievement in % | Judgment |
|---|---|---|---|---|---|
| Availability | 30 | 30 | 18.6 | 62.1% | Fair |
| Compliance | 40 | 40 | 30.2 | 75.5% | Good |
| Satisfaction | 30 | 30 | 14.1 | 46.9% | Poor |
| **Overall implementation** | **100** | **100** | **62.9** | **61.5%** | **Fair** |

3000 ETB were 154 (32.3%) and from them 64.3% were satisfied by the FP service. Only (44.4%) clients were satisfied with health facilities working hour. This was incongruent with the study conducted in Jimma zone which shows (97%) of the respondents were satisfied by the clinics working hour [16]. This discrepancy might be due to variation in the study period, study area and culture of the community. Most clients 298 (62.5%) was discussed on FP with their husbands and among them 49.7% were satisfied with the FP service they received.

The overall satisfaction of clients on FP service was 46.9. This finding was supported with a study conducted in Jigjiga town (41.7%) [33]. However, this finding is lower than studies conducted in Zambia (93%) [37], Iran (83.3%) [38], and Jimma zone (93.7%) [28]. This greater variation may be happened with due to variation in study period, expectation of a family planning user and cultural difference.

## Limitations of the evaluation

➢ Since the evaluation was facility based, it is possible that dissatisfied clients may not come to the health facilities. So, satisfaction result might be overestimated.

➢ Evaluating FP with only three evaluation dimensions may not be fair.

➢ Since it is cross-sectional evaluation, it doesn't show cause and effect relation, variation across areas and over time.

## Conclusion

Availability of resources for family service was fair while the compliance of healthcare providers was good by the presetting judgment matrix. But acceptability of family planning services by clients was judged as poor. Generally, the overall evaluation result of family planning program at Gondar city public health facilities in the three dimensions was fair based on pre-setting judgment criteria. Discussion with husband about family planning and waiting time to get the provider showed statistically significant association with client satisfaction of FP service. Therefore, health managers at all levels should fulfill all necessary infrastructures, supplies, IEC materials, and equipment for FP service and health care providers should comply the national FP service guideline. Further, the health facility managers should develop a strategy to empower women (non-pregnant women conference) and reduce wait time to get the service. Since contraceptives are basic inputs for family planning program, all the assigned health care providers should check the availability of necessary contraceptives and fill request form before stocked out these contraceptives.

## Supporting information

**S1 File. English version data collection tools.**
(PDF)

**S2 File. SPSS compliance dimension data.**
(SAV)

**S3 File. SPSS satisfaction dimension data.**
(SAV)

## Acknowledgments

We would like to thank University of Gondar, Gondar city health office and all study participants for their participation, information sharing and commitment. Our appreciation extended to the data collectors for their unreserved contribution.

## Declarations

**Ethics approval and consent to participate.** Ethical approval was obtained from the University of Gondar College of Medical and Health Sciences Ethical Review Committee with Ref No. IPH/837/6/2020. Data collection was undertaken after permission is obtained from Gondar city administrative health office. Written informed consent was taken from each participants. For clients age <18 years consent was taken from their parents or guardians.

## Author Contributions

**Conceptualization:** Tsega Hagos, Ayal Debie.

**Data curation:** Endalkachew Dellie.

**Formal analysis:** Getachew Teshale.

**Investigation:** Sefiw Abay.

**Methodology:** Ayal Debie.

**Validation:** Endalkachew Dellie, Lake Yazachew.

**Writing – review & editing:** Lake Yazachew, Getachew Teshale.

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
