## [Decision Letter · Decision Letter 0]

13 Jun 2022

PONE-D-21-38781Evaluation of Family Planning Service delivery in Gondar city public health facilities Northwest Ethiopia. Cross-sectional evaluation designPLOS ONE

Dear Dr. Teshale,

Thank you for submitting your manuscript to PLOS ONE. After careful consideration, we feel that it has merit but does not fully meet PLOS ONE’s publication criteria as it currently stands. Therefore, we invite you to submit a revised version of the manuscript that addresses the points raised during the review process.

We look forward to receiving your revised manuscript.

Kind regards,

Dylan A Mordaunt, MD, MPH, FRACP

Academic Editor

PLOS ONE

**Journal requirements:**

“The University of Gondar sponsored this evaluation. However, it has no role in the decision to publish, manuscript preparation, and publication.”

“The author(s) received no funding for this work.”

“The authors have declared that no competing interest.”

**Additional Editor Comments:**

Thank you for your submission. This is a mixed methods health services evaluation study. There appears to be good evaluation and data in this study, however there are significant issues with the way in which this is described, the writing and the resulting manuscript. The first issue I see is a lack of a clear description of the program/intervention that was evaluation, the context of that program, how it is delivred and how that may compare with similar family planning programs. One of the reviewers provides detail around this. From there it's largely about following reporting guidelines, clarity and fluency of the writing. Unfortunately there isn't a specific mixed methods reporting guideline or checklist that I'm aware of. There are some elements of the CHEERS economic evaluation checklist that would be useful (https://www.equator-network.org/?post_type=eq_guidelines&eq_guidelines_study_design=economic-evaluations&eq_guidelines_clinical_specialty=0&eq_guidelines_report_section=0&s=+), along with these papers- https://pubmed.ncbi.nlm.nih.gov/18416914/ and https://pubmed.ncbi.nlm.nih.gov/34253078/.

With specific regards to the criteria for publication:

1. The study appears to present the results of original research.

2. Results reported do not appear to have been published elsewhere.

3. Experiments, statistics, and other analyses are not clearly and completely described.

4. Conclusions are presented in an appropriate fashion and appear to be supported by the data.

5. The article is presented is reasonable to follow, though deficiencies in fluency impact this.

6. An IRB statement is present.

7. The article could be improved by utilising elements of structured reporting tools/guidelines.

Once these issues are addressed, this is likely to meet the criteria for publication. I look forward to receiving your resubmission.

Reviewers' comments:

Reviewer's Responses to Questions

**Comments to the Author**

1. Is the manuscript technically sound, and do the data support the conclusions?

Reviewer #1: Yes

Reviewer #2: Partly

Reviewer #3: Partly

2. Has the statistical analysis been performed appropriately and rigorously? 

Reviewer #1: Yes

Reviewer #2: No

Reviewer #3: No

3. Have the authors made all data underlying the findings in their manuscript fully available?

Reviewer #1: Yes

Reviewer #2: Yes

Reviewer #3: No

4. Is the manuscript presented in an intelligible fashion and written in standard English?

Reviewer #1: Yes

Reviewer #2: Yes

Reviewer #3: No

5. Review Comments to the Author

Reviewer #1: The study is well designed and the work process is carefully described. The results of this study are very valuable for evaluating the family planning program in the area; but it is clear that the results will be specific to the same time and place.

Reviewer #2: METHODS

- Please describe more clearly about the analysis of qualitative study in the method section.

RESULTS

- Please check table 4 – socio-demographic characteristics of clients using family planning services in Gondar City. The total frequency of each variable should be the same number. If it is different, please explain it in the method section.

- Please add more transcript of qualitative study. In the study, there are only 3 transcripts of the interviews.

- What is star (*) for in the table 6, please add in the note under the table.

- Table 6: why do you keep including variables that are not significant in bivariate analysis into multivariate analysis?

Discussion:

- Please add the recommendation related to the contraceptive stocked out in the health facilities in the past three months

Reviewer #3: Thank you for the opportunity to review this paper evaluating family planning service delivery in Gondar city Ethiopia. I have a number of issues with the manuscript in its current form

Major

1. Abstract-- The results reported in the abstract is limited in comparison to what was detailed in the methods section of the abstract.

2. Introduction – The introduction as is written is sometimes unclear to me, and the aim of the study appears vague. I suggest re-writing aspects of the introduction to improve clarity of the background behind the study and to clearly state the study aim/objective.

3. Methods—Lacking in specific details

a. Sampling: Can the authors provide details on what the sample distribution was within the eight family planning service facilities. i.e. did most of the sample come from just one of the facilities or was it evenly spread?

b. Variables (lines 149 to 155): Can the authors describe why they sought these independent variables? Based on experience or literature? Also the authors should state how they collected data for these variables – patient reported, interview or structured interview questionnaire?

c. It is not clear what was adjusted for in the multivariable logistic model, authors should also provide global p values for the regression on Table 6.

d. Can the authors please name and describe the validated tools that were adapted or used for assessing ‘availability of resources’, ‘Compliance’, and ‘Acceptability’?

e. Can you clarify what you mean by ‘formative approach’ on line 123

f. Can you clarify what you mean by ‘exit interview’ and ‘key informant interview’ on line 145, including which study participants where sampled for the respective interviews (i.e service users, program managers, healthcare workers etc.)

4. Results

a. In Table 3,5,& 7 the ‘Expected’ column is blank

Minor

1. I suggest the authors consider reviewing the title of their study to something along the lines, 'An evaluation of family planning service delivery in Gondar city public health facilities Northwest Ethiopia; a cross-sectional study'

2. Study dates don’t match up can you clarify. In the abstract you report ‘March 10 to April 25, 2020’ and on line 113 you report ‘March 25 to April 25, 2020’.

3. I am not sure ‘predictors’ (line 28) of an outcome can be assessed from a cross-sectional study. I suggest the authors rephrase this to ‘factors associated with patient satisfaction’

4. Manuscript needs proofing to correct some grammatical errors. For example in lines 54-56, lines 174, lines 223-225

5. Please define what EDHS stands for on line 100.

6. Also the authors had 100% response rate for the service user interviews. Does this mean all service users approached agreed to participate in the study?

6. PLOS authors have the option to publish the peer review history of their article (what does this mean?). If published, this will include your full peer review and any attached files.

Reviewer #1: **Yes: **Seyed Ali Azin

Reviewer #2: No

Reviewer #3: No

---

## [Author Response · Author response to Decision Letter 0]

19 Aug 2022

Of course; all service users asked to participate in the interview were agreed to respond the questionnaires.

---

## [Editor Report · Decision Letter 1]

23 Aug 2022

Evaluation of family planning service delivery in Gondar city public health facilities, Northwest Ethiopia: a cross-sectional study.

PONE-D-21-38781R1

Dear Dr. Teshale,

We’re pleased to inform you that your manuscript has been judged scientifically suitable for publication and will be formally accepted for publication once it meets all outstanding technical requirements.

Kind regards,

Dylan A Mordaunt, MD, MPH, FRACP

Academic Editor

PLOS ONE

Additional Editor Comments (optional):

Thank you for your resubmission. This now meets the criteria for publication.
---

## [Editor Report · Acceptance letter]

2 Sep 2022

PONE-D-21-38781R1 

Evaluation of family planning service delivery in Gondar city public health facilities, Northwest Ethiopia: a cross-sectional study. 

Dear Dr. Teshale:

I'm pleased to inform you that your manuscript has been deemed suitable for publication in PLOS ONE. Congratulations! Your manuscript is now with our production department. 

Kind regards, 

on behalf of

Associate Professor Dylan A Mordaunt 

Academic Editor

PLOS ONE